# On the Stability of Electrohydrodynamic Jet Printing Using Poly(ethylene oxide) Solvent-Based Inks

**DOI:** 10.3390/nano14030273

**Published:** 2024-01-27

**Authors:** Alberto Ramon, Ievgenii Liashenko, Joan Rosell-Llompart, Andreu Cabot

**Affiliations:** 1Catalonia Institute for Energy Research (IREC), Jardins de les Dones de Negre 1, Sant Adrià de Besòs, 08930 Barcelona, Spain; bertoramf@gmail.com (A.R.); liashenk@uoregon.edu (I.L.); 2Department of Chemical Engineering, University Rovira i Virgili, Av. dels Països Catalans 26, 43007 Tarragona, Spain; 3Phil and Penny Knight Campus for Accelerating Scientific Impact, University of Oregon, 1505 Franklin Boulevard, Eugene, OR 97403, USA; 4Catalan Institution for Research and Advanced Studies (ICREA), Pg. Lluís Companys 23, 08010 Barcelona, Spain

**Keywords:** electrohydrodynamic jet printing, near-field electrospinning, fiber, 3D printing

## Abstract

Electrohydrodynamic (EHD) jet printing of solvent-based inks or melts allows for the producing of polymeric fiber-based two- and three-dimensional structures with sub-micrometer features, with or without conductive nanoparticles or functional materials. While solvent-based inks possess great material versatility, the stability of the EHD jetting process using such inks remains a major challenge that must be overcome before this technology can be deployed beyond research laboratories. Herein, we study the parameters that affect the stability of the EHD jet printing of polyethylene oxide (PEO) patterns using solvent-based inks. To gain insights into the evolution of the printing process, we simultaneously monitor the drop size, the jet ejection point, and the jet speed, determined by superimposing a periodic electrostatic deflection. We observe printing instabilities to be associated with changes in drop size and composition and in the jet’s ejection point and speed, which are related to the evaporation of the solvent and the resulting drying of the drop surface. Thus, stabilizing the printing process and, particularly, the drop size and its surface composition require minimizing or controlling the solvent evaporation rate from the drop surface by using appropriate solvents and by controlling the printing ambient. For stable printing and improved jet stability, it is essential to use polymers with a high molecular weight and select solvents that slow down the surface drying of the droplets. Additionally, adjusting the needle voltages is crucial to prevent instabilities in the jet ejection mode. Although this study primarily utilized PEO, the general trends observed are applicable to other polymers that exhibit similar interactions between solvent and polymer.

## 1. Introduction

Electrohydrodynamic (EHD) jet printing, also known as near-field electrospinning (NFES), is a nano/micro-fabrication technology for producing fiber-based structures and devices directly and continuously [1,2,3]. This technology is based on creating an electric field between a conductive nozzle tip (or “needle”) and a collector to generate electrostatic forces on the surface of an ink drop at the nozzle exit. For a sufficiently large voltage difference between the nozzle and the collector substrate, a thin filament of ink, or jet, is ejected from the ink drop towards the substrate, where micro/nano-fibers are collected [4,5,6,7,8,9]. This approach allows the precise deposition of fibers, with diameters down to the nanometer scale regime, on the collector to print two- and three-dimensional structures with sub-micrometer features [10,11,12,13,14].

Owing to its large material versatility, this technology has been used for the production of a plethora of materials for a wide range of applications, from printed electronics [15] to batteries [16,17] and tissue engineering [18], among others [19]. Recently, breakthrough progress in the electrostatic control of the jet’s positioning during its travel to the collector has further pushed the potential of this technology for commercial applications [20,21,22,23,24]. In particular, the use of jet-deflecting electrodes allows ultrafast printing by electrostatically stirring the EHD jet [20]. However, while EHD jet printing is potentially a cost-effective technology, the jet’s stability is very sensitive to a number of parameters, especially when using solvent-based inks (in contrast to melts). Such a sensitivity often limits the printing stability and reproducibility [25,26]. Therefore, to ensure reliable and reproducible printing, it is critical to either avoid instabilities or detect and correct them in situ. Both of these tasks require knowing the origin of those instabilities.

Some previous works have demonstrated the influence of specific parameters on jet speed, such as the needle voltage, the infusion pump rate, the nozzle-to-collector distance, or the polymer concentration [27,28,29,30,31,32,33,34]. However, the current understanding of how jet stability is affected by ink composition, processing parameters, and ambient conditions is still rather poor. This is true even for the “work-horse” solution inks containing polyethylene oxide (PEO) as a solute, on which much fundamental development has been based. Concerning solvent-based inks, the addition of high-molecular-weight polymers to the ink to increase its viscoelasticity has proven to be effective in preventing the breaking of the EHD jet, permitting a more stable printing process [35,36,37,38]. However, we find that the deposition of fiber-based patterns on a substrate using solvent-based inks is prone to additional instabilities associated with solvent evaporation from the drop and the jet. In this regard, the effects of the drop size and the solvent evaporation rate on the stability of the process and on the properties of the printed fibers have generally been overlooked.

Here, we identify several instabilities that occur during the printing process, analyze how they are influenced by different printing process parameters, and define strategies to avoid them. We focus on solution-based inks containing PEO of different molecular weights for their continued importance and assumed stability in the literature about EHD spinning processes. To analyze the stability of our printing process, we not only examine the printed patterns but also monitor the pendant drop, the jet ejection point, and jet speed in real time. To determine the jet dynamics, the jet is electrostatically deflected using periodic signals [20]. This approach allows us both to control the jet trajectory to print any desired pattern and to simultaneously estimate the jet speed from the width of the printed pattern [21,22].

This article’s results are logically structured as follows. Initially, in Section 3.1, we study the stability of the jetting for different polymer molecular weights in water:ethanol solutions of similar viscosities and confirm the benefit of increasing the molecular weight of the polymers to prevent jet interruptions. This step allows us to select a polymer molecular weight (of 1 MDa) for the rest of our analyses. Next, in Section 3.2, the time evolution of the drop size is examined alongside the “microscopic” variables fiber diameter and jet speed, the latter being determined using the jet deflection method. The jet’s disappearance can be triggered by exceedingly low infusion flow rates, even after the drop volume has stabilized. Next, in Section 3.3, we describe a transient jet pulsation regime for an excessive nozzle voltage, which may cause an imbalance between the depletion and supply rates in the jet. Finally, in Section 3.4, we describe the wandering of the printing jet that results from the unsteady drifting of the jet ejection point on the drop and show that this instability can be suppressed by reducing the drying rate of the drop.

## 2. Materials and Methods

### 2.1. Chemicals

Poly(ethylene oxide) of various viscosity-average molecular weights were purchased from Sigma-Aldrich (San Luis, MO, USA) (#182001, 0.3 MDa; #182028, 0.6 MDa; #372781, 1 MDa; #189472, 5 MDa). Ethanol (EtOH) and ethylene glycol (EG) were obtained from different sources. Reagent-grade deionized water was obtained from Purelab flex, by Elga (Lane End, UK). All the chemicals were used as received, without further purification.

### 2.2. Ink Formulation

Inks were prepared by dissolving a proper amount of PEO (between 1 and 6 wt%, depending on its molecular weight) in an aqueous solvent mixture either containing EtOH or EG to tune the surface tension and evaporation rate. PEO dissolution was carried out by mixing the solution for 24 h under magnetic stirring. The ink compositions and properties are given in Appendix A. Ink electrical conductivity and viscosity were measured using a water hardness sensor (Lutron Electronic (Taipei, Taiwan), model CD–4309) and a viscometer (Brookfield Engineering (Middleboro, MA, USA), model DV2T, with spindles SC4-18 and SC4-34).

### 2.3. Printer Set-Up and Printing Protocol

Figure 1 illustrates the device used for EHD jet printing. The ink was loaded into a glass syringe (Hamilton (Reno, NV, USA) #81320, 1 mL) and supplied to the nozzle using a syringe pump (Harvard Apparatus (Cambridge, MA, USA), Pump 11 Pico Plus Elite 70-4506) with infusion rates in the range 20–70 nL min^−1^. Stainless steel needles with a blunt ending (Hamilton N726S, 26 s gauge, 127 μm ID, 474 μm OD) and borosilicate glass tips (ca. 200–300 µm OD) were used as the nozzles. The glass tips were produced from borosilicate glass capillaries (Sutter Instruments (Novato, CA, USA), B100-50-15) using a pipette puller (Sutter Instruments P-97) and were inserted and glued to a hypodermic needle. A positive potential was applied to the needle using a Matsusada AU-20P15 high-voltage power supply. Silicon wafers (University Wafers (Boston, MA, USA) #452, p-type, <100>) were used as substrates and were placed on an electrically grounded plate mounted atop a motorized XY translation stage (PI miCos (Karlsruhe, Germany) linear stages PLS-85). The conductive nature of the substrate was intended to help dissipate the electrostatic charge associated with the deposited material. The nozzle-to-substrate distance was fixed at 3 mm.

The EHD jet was initiated by applying a voltage in the range 800–1200 V on the needle and by piercing the pendant drop with a hypodermic needle. (This protocol was applied also to reinitiate the jetting after spontaneous stopping.) The voltage required for jet initiation was lower for needles and pendant drops of a smaller size. Once the jet was initiated (Time = 0), it was initially printed for 2 min on the wafer substrate moving slowly for stabilization (at 0.175 mm s^−1^). To determine the effect of different parameters on the printed patterns, the stage translated the substrate in a zigzag pattern made of 10 mm long translations along the X-axis, with 0.5 mm displacements along the Y-axis in-between, until a printing area of 10 × 10 mm^2^ was covered (Appendix A). The linear speed of the stage was raised to 1 mm s^−1^ following the 2 min stabilization period. This speed was much below the speed of the jet. Additionally, the translation path of the stage was used to analyze the deviation of the default jet trajectory over time. Appendix A explains the procedure followed to analyze the jet deviation from its default trajectory during printing.

The jet was deflected from its default trajectory using two orthogonally arranged jet-deflecting electrodes. Such auxiliary electrodes were located 10 mm away from the needle axis and are described in a previous publication [20]. A data acquisition card (National Instruments (Austin, TX, USA), USB-6259) was used to generate the analog signals (max. ±10 V), which, after amplification (Matsusada (Shiga, Japan) AMJ-2B10 and Trek 677B), were applied to the jet-deflecting electrodes. The signals were synchronized sinusoidal functions of time, with a 2000 V amplitude and a frequency selected in the range 40–500 Hz.

During the printing process, the ink drop at the end of the nozzle tip was continuously monitored at five frames per second using a CMOS camera (Basler (Ahrensburg, Germany) acA2040-25gc) mounted on a microscope consisting of a 12X lens with adjustable zoom and focus (Navitar (Rochester, NY, USA) 1-50486), a 2X lens adaptor (Navitar 1-62136), and a 5X microscope lens (Mitutoyo (Kawasaki, Japan) 1-60226), resulting in a nominal working distance of 34 mm. A high-intensity light source (AmScope (Irvine, CA, USA) HL-250-A) was positioned behind the observed object at roughly a 5° angle from the camera’s optical axis to minimize the source’s light entering the optics. To produce bright-field illumination, the light was aimed at a paper sheet placed between the drop and the light source, which provided a lit background in the images.

The printing process was carried out under room ambient conditions, at a temperature and relative humidity in the ranges 18–19 °C and 40–60%, respectively. To ensure reproducible solvent evaporation, a flow of dry nitrogen gas (700 mL min^−1^) was supplied around the jet from a tube located next to the nozzle. During printing, the substrate was continuously translated relative to the nozzle using the motorized XY stage. The detailed printing parameters are given in Table 1 and Appendix A.

### 2.4. Drop Size and Volume Computation

During the printing process, the ink drop radius and volume were estimated from the CMOS camera frames, as illustrated in Figure 1. First, an axisymmetric coordinate system was defined at the drop by fixing its origin at the jet ejection spot and dividing the drop into ∆Z portions. An XY digitizer provided the coordinates of the drop edge on the captured image, which were used to compute the partial volume of the drop (${∆V}_{i}$) as follows:(1)$${∆V}_{i}=\frac{1}{3}\cdot \pi \cdot ∆Z\cdot \left(R_{i}^{2}+R_{i+1}^{2}+R_{i}\cdot R_{i+1}\right)$$
where ∆Z is the distance between two different points along the drop in “*Z*” axis, and $R_{i}$ is the drop radius (Figure 1). Then, the total drop volume ($V_{d}$) was calculated as follows:(2)$$V_{d}=2\cdot \left(\sum_{i=1}^{n}{∆V}_{i}\right)$$

### 2.5. Fiber Characterization

The diameter of the fibers was imaged using an Auriga scanning electron microscope (SEM) from Carl Zeiss operated at 1–5 kV and using an in-lens detector. Previously, the samples had been sputtered with a thin copper layer using a DC magnetron sputter (Emitech (Montigny-le-Bretonneux, GFrance) K575X, 85 mA, argon, 90 s) to improve the quality of the SEM images and to protect the PEO fiber from the degradation/shrinkage caused by the electron beam.

### 2.6. Jet Speed and Flow Rate Determination

To determine the jet speed, the jet was cyclically deflected electrostatically along the Y-axis, orthogonally, to the direction of displacement of the stage (X-axis). After printing, the width of the printed track was measured with a confocal microscope (Sensofar PLu Neox (Barcelona, Spain)). Polarized light was used to improve the contrast between the printed fibers and the silicon substrate. From the measured width of the fiber track (W) and the known frequency of the jet-deflecting signal (ν), the jet speed at its arrival to the substrate ($U_{Jet}$) was computed as follows [21]:(3)$$U_{Jet}=2W\nu$$

This expression is applicable when $U_{Jet}$ is much higher than the stage translation speed.

The rate of the printed polymer transported by the jet may not coincide with the infused polymer flow rate. To determine the former, two assumptions were made: (i) fibers have a circular cross-section and are not porous, so their volume coincides with the volume of printed polymer and can be computed from the fiber diameter (D) and W; and (ii) the ink density ($\rho_{Ink}$) can be computed from the known weights and densities of the constituent polymer and solvents. Thus, the polymer volumetric flow rate ($Q_{P,jet}$) propelled in the jet can be estimated as follows:(4)$$Q_{P,jet}=U_{Jet}\cdot A=\frac{\pi }{2}W\nu D^{2}$$
where A is the cross-section area of the printed fiber.

Then, considering the polymer density ($\rho_{P}$), the polymer mass flow rate in the jet ($M_{P,\;jet}$) can be computed as follows:(5)$$M_{P,jet}=\frac{\pi }{2}W\nu D^{2}\rho_{P}$$

Finally, by assuming that the weight concentration of the polymer in the jet is equal to the initial solution value ($C_{P,\;ink}$) (i.e., in the absence of solvent evaporation), the upper bounds to the jet mass flow rate ($M_{Jet,\;upper}$) and the jet volumetric flow rate ($Q_{Jet,\;upper}$) can be estimated:(6)$$M_{Jet,\;upper}=\frac{M_{P,jet}}{C_{P,ink}}$$
(7)$$Q_{Jet,upper}=\frac{M_{Jet}}{\rho_{Ink}}$$

### 2.7. Jet Deviation from Its Default Jet Trajectory

The jet trajectory was analyzed by monitoring deviations from the centerline of the printed patterns (Appendix A). To quantitatively measure the jet deviation, first, a default jet trajectory was specified. Its starting point was placed at the location where the jet was initially ejected stably to start the experiment. Then, its default trajectory was defined as that generated from that location by the stage pathway. As shown in Appendix A, the deviation from the default jet trajectory was measured as the distance between the fiber track centerline and the default jet trajectory.

## 3. Results and Discussion

Figure 2a displays a thorough list of the parameters that govern the EHD printing process and that define the properties of the printed product. The list is classified into operational (or control) variables, dependent (or response) process variables, and the characteristics of the printed pattern. To rationalize the complex EHD printing process, the effect of the dependent variables on the printing stability and performance is investigated by monitoring the drop size, the jet ejection point, and the patterns printed by electrostatically deflecting the jet (Figure 2b). We start by analyzing the influence of the ink parameters, particularly the polymer molecular weight and concentration and the related ink viscosity and electrical conductivity, as they determine the operational values of the process variables.

### 3.1. Influence of the Polymer Molecular Weight

When formulating an ink, it becomes necessary to increase its viscosity to enhance the stability of the EHD jet against the development of capillary waves, which lead to the Plateau–Rayleigh instability and can result in a beaded-fiber formation or even in the breaking up of the jet into droplets. In our jets, the presence of a polymer in the ink is, therefore, essential to ensuring a stable jet, and its molecular weight influences the printing process in several ways [36].

At a set polymer mass concentration, increasing its molecular weight results in an increase in the ink’s viscosity as a consequence of a larger entanglement number. Thus, to determine the influence of the polymer’s molecular weight independently from the obvious effects of Newtonian viscosity, we produced jets from inks containing different concentrations of PEO with distinct molecular weights, in the range from 0.3 MDa to 5 MDa, such that they had a similar zero-shear viscosity of around 1.8–1.9 Pa s (Appendix A). Obtaining similar ink viscosities with increasing the polymer’s molecular weight involves decreasing the polymer concentration and, consequently, in our case, also the ink’s electrical conductivity (Appendix A), which is expected to also affect the jet speed [39,40]. The jet speed was determined by measuring the width of the track printed when cyclically deflecting the jet using auxiliary electrodes, as shown in Figure 3 and Figure 4. Notice that, to obtain the patterns displayed in Figure 3c and, from it, the jet speed (Figure 4), the frequency of the deflecting signal had to be adapted to produce reproducible fiber patterns that allow for the measuring of the jet speed (i.e., patterns with straight fibers) [21]. At lower PEO molecular weights, too low frequencies resulted in buckled fibers associated with faster jets, while, at high PEO molecular weights, the slow jet could not follow deflecting signals of too-high frequencies, resulting in uncontrolled fiber patterns.

As can be observed in Figure 4, increasing the polymer molecular weight results in a marked reduction in the jet speed. Such trend is associated with both a reduction in the ink’s electrical conductivity [11,30,41,42] and an increase in the ink’s elasticity (i.e., higher entanglement number, Appendix A). Therefore, the jet is slowed down by stronger elastic forces that work against the jet stretching as well as by reduced electrical forces associated with less charge density when the electrical conductivity is reduced.

Additionally, Figure 4 and Appendix A display the time evolution of the jet speed while printing the patterns in Figure 3c, showing that, during the 5 min of printing, the jet spontaneously stopped twice when using ink containing 0.3 MDa PEO, just once when using 0.6 MDa PEO, and it did not stop for 1 and 5 MDa PEO, for which the printing was stopped intentionally at a time of 5 min. The difference in stability between the 0.6 MDa and 1 MDa solutions is remarkable. This is shown by subsequent experiments (shown later), in which we were able to continuously print 1 MDa PEO for more than 30 min. The greater stability enabled by the higher-molecular-weight PEO may be related to the higher entanglement number of the polymer chains in the solution (Appendix A), which increases the ink’s elasticity in the jet. Such an increase in elasticity stabilizes the jet against breaking while it is being stretched by the electrical field [35,43,44]. Moreover, the use of lower concentrations of high-molecular-weight polymers to maintain the same ink viscosity reduces the extent of drop surface drying, additionally increasing printing stability.

While the relative (normalized) decay rate of the jet speed was nearly independent of the polymer molecular weight, as shown in Appendix A, the decrease in the jet speed over time was more evident for the low-molecular-weight PEOs. These variations in the jet speed were clearly visible in the printed patterns only when using low-molecular-weight PEOs. For the low-molecular-weight PEO (0.3 MDa), variable track widths and, thus, fiber lengths were obtained along the track, demonstrating an unstable jet speed (Figure 3c, top image). In contrast, relatively uniform fiber track widths were obtained with the slower jet from high-molecular-weight PEOs (Figure 3c, bottom image). Considering that the 1 MDa ink met a good compromise of a higher jet speed and good stability during printing, it was chosen for further experiments on the stability of the printing process.

### 3.2. Stability of the Drop Size

A key observable parameter affecting and providing information on the stability of the printing process is the pendant drop size. The pendant drop at the exit of the capillary modifies the electric field between the nozzle and the collection substrate, thus altering the electric pull on the jet toward the collector and, consequently, influencing its speed. On the one hand, the drop introduces electrostatic shielding at the jet’s emission point, thus weakening the electrical field in that region. In other words, the larger the drop size, the greater the shielding, and, thus, the lower the electric field at the drop tip. On the other hand, at a set nozzle-to-substrate distance, an increase in the drop size reduces the distance between the drop’s surface and the substrate, so the electrical field at the drop tip is enhanced. Overall, in a change in drop size, the electrical field distribution around the jet changes and so does the speed at which it is electrostatically propelled toward the substrate. Thus, one a priori aims to adjust the infusion pump rate so that the drop size stays constant.

Figure 5 shows the time evolution of the drop size, jet speed, jet volumetric flow rate, and fiber diameter at two infusion pump rates, 20 and 40 nL min^−1^. While the smallest infusion pump rate allows for reaching a stable drop size sooner, the highest infusion pump rate results in a sustained increase in the drop dimensions (Figure 5a and Appendix A). To compare the experimental evolution in the drop size with the theoretical evolution, we applied a mass balance around the drop considering the amount of mass that enters and exits the drop, without considering mass accumulation or mass generation. Per unit time, the volume of ink pumped into the drop ($Q_{Ink}$) must equal the sum of the volume of ink ejected through the jet ($Q_{Jet,\;upper}$), the volume of solvent evaporated from the drop ($Q_{E}$), and the volume change in the drop ($\frac{dV_{d}}{dt}$):(8)$$Q_{Ink}=Q_{Jet,upper}+Q_{E}+\frac{dV_{d}}{dt}$$

Assuming that the solvent evaporation rate from the drop evolves with the variation in the drop radius only [45], i.e., the ink’s composition and the ambient temperature and pressure remain constant, then the following is fulfilled:(9)$$Q_{E}=B\cdot R$$
where B is a constant parameter. Then, Equation (8) can be expressed as follows:(10)$$\frac{dR}{dt}=\frac{Q_{Ink}-Q_{Jet,upper}}{4\pi R^{2}}-\frac{B}{4\pi R}$$

Equation (10) needs to be solved numerically. Considering the two $Q_{Ink}$ experimentally tested, 40 and 20 nL min^−1^, the average value of $Q_{Jet,\;upper}$ measured for each infusion pump rate (4.8 and 2.4 nL min^−1^, respectively), and the same constant B for both rates of 0.0065 mm^2^ min^−1^, the theoretical trend properly reproduces the observed evolution of the drop radius (Figure 5a and Appendix A).

On the other hand, we notice that the high infusion pump rate condition provides a more stable jet speed and, thus, more stable printing (Figure 5b). At the low infusion pump rate (the smallest tested in this work), both the jet speed and jet flow rate strongly decrease shortly after the drop size has stabilized, to finally disappear after around 15 min of jetting (Figure 5b,d). At both flow rates tested, the fiber diameter is remarkably (but maybe coincidentally) similar, and it remains fairly constant over time, although it tends to increase (Figure 5c).

Experimental conditions in row B of Table 1. A more realistic analysis of the process should consider the fact that, due to solvent evaporation, the polymer concentration within the drop ($C_{P,\;drop}$) continuously increases, thus affecting the drop properties. Assuming that the polymer concentration of the jet (at the ejection point) is the same as the average polymer concentration within the drop, the polymer concentration should evolve in time as follows:(11)$$\frac{dC_{P,drop}}{dt}=\frac{M_{P,ink}-M_{Jet,upper}}{V_{d}}=\frac{Q_{Ink}C_{P,ink}-Q_{Jet,upper}C_{P,drop}}{V_{d}}$$
where $M_{P,\;ink}$ is the polymer mass flow rate pumped into the drop. Assuming $Q_{Jet,\;upper}$ to be constant (Figure 5d) and considering that the volume of the drop changes with time, the following is observed:(12)$$C_{P,drop}=C_{P,drop\;0}+\frac{C_{P,ink}\left(Q_{Ink}-Q_{Jet,upper}\right)}{Q_{Jet,upper}}e^{-Q_{Jet,upper}\int \frac{1}{V_{d}}dt}$$

Thus, the polymer concentration increases at a rate that depends on the difference between the infusion pump and the jet’s volumetric flow rates and the rate of volume change. Only when considering a homogenous polymer concentration within the ink, a constant drop volume, and no solvent evaporation, $Q_{Ink}$ − $Q_{Jet,\;upper}$ = 0, and, then, $C_{P,drop}$ should be constant, not perturbing the printing process.

Since the ink supply takes place at the nozzle end and the solvent evaporation occurs on the surface of the drop, a polymer concentration gradient inevitably exists within the drop, where probably the real polymer concentration at the drop surface is much larger than the average polymer concentration within the drop. Such polymer enrichment is accentuated by the slow diffusion of the polymer. Over time, this polymer enrichment increases the ink viscosity on the drop surface and, therefore, in the jet, which decreases the jet speed and increases its thickness. Such trends are observed in Figure 5b,c, although they are slow. Also shown in Figure 5b,d is that, to decrease the rate of surface polymer enrichment, a more stable printing is obtained using infusion pump rates that are significantly larger than the jet ejection rates; thus, the drop size generally increases (Figure 5a).

Because the volume of the solvent evaporated increases with the drop radius, at some point the drop size is stabilized (Figure 5a). When the drop size becomes stable, i.e., its volume remains constant, a significant portion of the ink’s solvent injected into the drop is evaporated from the surface of the drop. As an example, at an infusion pump rate of 20 nL min^−1^, once a stable drop size is reached, the estimated jet flow rate is ~3 nL min^−1^ (Figure 5d); thus, the solvent evaporation rate is ~17 nL min^−1^. In this scenario, the average polymer concentration within the drop, particularly on its surface, must be much higher than in the original ink.

The jet viscosity is also increased, resulting in a sustained reduction in the jet speed, as the viscous forces work against the jet flow, even to the point that it can disappear, as is found in the case of 20 nL min^−1^. Higher infusion pump rates better stabilize the printing process against the drying of the drop surface, as the effect of solvent evaporation is less noticeable because the drop growth and the refreshment of ink on the drop surface slow down the surface polymer enrichment. However, as the drop size increases, the jet speed may be affected by the perturbations in the electric field around the jet. Thus, excessive infusion pump rates destabilize EHD printing and have to be avoided as well.

### 3.3. Pulsating Jet Transient Regime

The needle voltage plays a key role in controlling the printing speed since higher needle voltages result in faster jets, thus increasing the $Q_{Jet}$ and the printing process speed. However, when too-high $Q_{Jet}/Q_{Ink}$ are used, an intrinsic pulsating jet ejection regime can be observed by visualizing the drop with the CMOS camera (Appendix A) and on the printed pattern (Figure 6 and Appendix A). In contrast to previous reports on the pulsation mode [46,47,48], we did not observe an on-off pulsation of the jet ejection but a localized pulsation of just the jet emission point on the drop (Appendix A).

Figure 6 and Appendix A show fibers printed at four needle voltages in the range 850–1150 V, where the jet was electrostatically deflected to monitor its speed. Each printing at a different nozzle voltage was an independent experiment (see the Experimental Section for details on drop initialization). Homogeneous fiber tracks were printed at 850 and 950 V, resulting in jet speeds of 55 and 87 mm s^−1^, respectively. At higher needle voltages, pulses in the jet emerged while the drop was growing (as in Figure 5a). At 1050 V, the pulsation revealed in the printed track width revealed a frequency of 3 Hz. (This, incidentally, was too fast to be quantified in our videos, taken at five frames per second.) The jet was much faster at this voltage than at the lower ones, as revealed by the fact that the printed fiber was significantly buckled. However, the track width varied drastically through the oscillation, indicating significant changes in the jet speed through each oscillation cycle. From the fiber track, we could infer upper bounds on the jet speed of 200 mm s^−1^ and 140 mm s^−1^ at the maximum and minimum of each oscillation. Eventually, the pulses disappeared, producing straight fibers at 75 mm s^−1^.

At an even higher needle voltage (1150 V), a similar but longer-lasting pulse mode was observed, also with buckled fibers and at a similar frequency of 3.5 Hz. The pulsation disappeared eventually (as with the 1050 V case) when the jet speed decreased probably because of the drying of the drop (Appendix A). To avoid this regime, high-enough infusion pump rates and low-enough needle voltages need to be used to obtain a proper $Q_{Jet}/Q_{Ink}$ ratio [48].

Pulsations of Taylor cones for Newtonian low viscosity fluids are typically encountered as needle voltage is decreased relative to the stable jetting condition (rather than increased) and result in the interruption of the jetting in each pulsation period [47]. The jet is interrupted in such conditions due to hydrodynamic and mechanical reasons. At low infusion pump rates (our $Q_{Ink}$), a chocked-flow pulsating regime is attained, where an imbalance exists between the higher rate at which the jet depletes the drop and the slower refill rate of the liquid supply through the nozzle. At higher infusion pump rates, the rate of liquid supply is high enough, and the jet oscillations reflect the natural mechanical oscillations of the liquid meniscus. In our case, the pulsation suggested also an imbalance between the depletion and supply liquid rates at the jet’s location, although the elastic stresses may have prevented the periodic collapse (disappearance) of the jet.

### 3.4. Unstable Default Jet Trajectory

We also analyzed the effects of the needle voltage and the infusion pump rate on the jet ejection point. We found that the location of this point usually changes the default jet trajectory from the drop toward the substrate. In these tests, we varied the needle voltage and infusion pump rate in the ranges of 850–1150 V and 30–70 nL min^−1^, respectively. These ranges were chosen because they allowed the generation of long-living jets and their printing as fibers on the substrate. Too-low needle voltages did not generate a jet, whereas too-high voltages produced too-fast jets that had not enough time to dry before reaching the substrate, where they formed ink puddles. Moreover, lower infusion pump rates resulted in the fast drying of the drop (Figure 5), while higher infusion pump rates resulted in fast drop growth which eventually perturbed too strongly the electrical field. Figure 7a displays a jet being ejected from a drop when the needle voltage increases from 850 to 1050 V at a fixed infusion pump rate of 70 nL min^−1^ (Appendix A displays the jet at those conditions and also when the needle voltage was increased to 1150 V). Figure 7b and Appendix A show the jet being ejected from the drop when the infusion pump rate decreases from 70 to 30 nL min^−1^ and the needle voltage is fixed at 850 V.

As observed in these figures and videos, increasing the needle voltage did not significantly perturb the jet ejection point. On the other hand, when the infusion pump rate was decreased, the drop size decreased, and the jet ejection point shifted sideways.

The use of jet deflection electrodes allowed for the in situ detection of instabilities in the position of the jet ejection point and the default jet trajectory as displacements of the track centerline (Figure 8a,b). As shown in Figure 8b, the jet ejection point displacement correlates with the default trajectory displacement, and such displacements take place without any appreciable change in the size or shape of the drop. Note that the anchoring of the drop on the capillary remains the same while the locations of the jet ejection point differ. Figure 8c displays confocal images of the printed tracks and the time evolution of their centerline deviation from the translation path of the stage. The default jet trajectory, or the track centerline, becomes unstable when reducing the infusion pump rate, which is consistent with the sideways shifting of the jet ejection point on the drop, which is also seen for this condition in Figure 7b and Appendix A.

The jet ejection point on the drop surface is determined not only by the electric field distribution but also by the drop surface composition. Therefore, we hypothesized that the unsteady shifting of the point of ejection of the jet could be related to compositional inhomogeneities on the surface of the drop that may develop while it dries. This hypothesis is consistent considering that the effect of the drying of the surface of the drop was more accentuated when using low infusion pump rates (Figure 5). Inhomogeneities on the surface composition of the drop, e.g., associated with irregular drop drying, translate into disparities in surface tension, which, via Marangoni stresses, could shift the jet ejection point and the default jet trajectory, thus resulting in a translation of the center of the printed pattern, as displayed in Figure 8c. Besides surface tension gradients, other mechanisms caused by inhomogeneities can be conceived, such as the reinforced increase in viscosity in a region where evaporation is slightly faster, leading to an increase in viscosity and, thus, a slower flow in and around the region, with the jet naturally shifting away from it, as the field continues to pull liquid from lower-viscosity regions nearby.

To assess whether these instabilities are connected to ink drying, we reduced the volatility of the ink by changing the solvent from EtOH to EG. With an ink composed of 3 wt% PEO (1 MDa) dissolved in H_2_O:EG (4:1 wt), the most significant changes in the ink properties, besides volatility, were the increases in both ink viscosity and electrical conductivity (Appendix A). To print dry fibers, the jet speed was reduced by decreasing the needle voltage. Figure 8d displays the deviation of the centerline of the printed fibers when the infusion pump rate was decreased from 70 to 30 nL min^−1^ at a constant needle voltage (700 V), revealing that, with the EG-based ink, the default jet trajectory and the jet ejection point remained stable even at the lowest infusion pump rates used. Appendix A shows that the jet being expelled from the drop under these conditions is centered and stable.

To further study the effect of solvent volatility on the printing process, Figure 9 displays the time evolution of the drop volume, centerline deviation of the printed track, jet speed, and fiber diameter during printing with the EG-based ink. The lower solvent evaporation rate associated with the use of EG instead of EtOH leads to more stable and faster jets, as well as thinner fibers, as a result of a lower drop viscosity due to the refreshment of its surface with a higher amount of fresh ink by unit of time. Thus, we conclude that controlling and limiting the drying of the drop surface and using polymers with high molecular weights are key to stabilizing the jet for a long time and printing reproducible patterns.

## 4. Conclusions

Steady-state printing requires the control and stabilization of several parameters. In EHD jet printing using solvent-based inks, the jet speed stability strongly depends on the inks’ properties. We observed for PEO-based inks that raising the molecular weight helped in preventing the interruption of the jet and ensured a more uniform jet speed, but it also resulted in the ejection of slower jets. Additionally, solvent evaporation from the drop must be minimized to avoid the drying of its surface, which strongly affects the drop properties and, thus, the jet dynamics. The ejection of the jet is influenced not only by the electric field distribution at the drop tip but also by the composition of its surface. On the one hand, strong electric fields result in a pulsation mode that leads to variable jet speed during printing. On the other hand, a volatile solvent, which promotes the drying of the drop surface, perturbs the ejection point of the jet, causing the printed pattern to drift. Therefore, to assure reproducible printing, high polymer molecular weights have to be used with solvents that minimize the drying of the drop surface, and needle voltage has to be optimized to avoid instabilities in the jet’s ejection mode. Finally, it should be noted that, while the results reported in this work were obtained using PEO as the printed material, the general tendencies identified can be extended to solvent-based inks of other polymers with similar solvent–polymer interactions.

## Data Availability

Original measurement data are available upon reasonable request.

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
