# Peer review of "On the Stability of Electrohydrodynamic Jet Printing Using Poly(ethylene oxide) Solvent-Based Inks"

_nanomaterials, 2024, doi:10.3390/nano14030273_

Round 1

Reviewer 1 Report

Comments and Suggestions for Authors

This manuscript experimentally investigates the parameters that affect the stability of EHD jet printing of poly (ethylene oxide) patterns using solvent-based inks, some guidance is provided for the development and improvement of steady-state printing. Although some results are obtained from experiment and calculation, there are still some issues, we suggest that it be published after major modifications. In addition, there are the following questions that the author should consider and revise:

1.       The introduction section needs to be enriched with previous research in EHD area.

2.       In section 2.3, we suggest that a picture of the device be listed, and that the device be analyzed in conjunction with the picture to facilitate the reader's understanding.

3.       The sources of all formulas in the manuscript need to be clarified.

4.       In section 3.1, the associations of polymer molecular weight with viscosity and conductivity are not clearly articulated. The authors also changed the viscosity when exploring the influence of the polymer molecular weight, and had problems controlling the variables.

5.       Table S1, table A.1… appears several times in the manuscript, but there is only one table 1 in the article.

6.       Figures 4b, c, d need to be revised and the working conditions represented by the black and white squares need to be identified in each figure.

7.       What the association of high polymer molecular weight with the drop surface drying? Authors need to elaborate clearly in the text.

8.       As we all know, the main factors affecting the atomization of conical jets are voltage, flow rate, conductivity and surface tension, why did the authors choose polymer molecular weight for their study?

9.       The logic of the article is confusing and needs to be adjusted to make it reader friendly.

10.   Please use updated and recent papers in the literature review to give more sense to the reader.

A conservative level set method for liquid-gas flows with application in liquid jet atomization. Experimental and Computational Multiphase Flow, 5, 67-83 (2023)

Experimental study on electrohydrodynamic atomization (EHDA) in stable cone-jet with middle viscous and low conductive liquid. Experimental Thermal and Fluid Science, 121, 110260.

Author Response

We would like to express our gratitude to the reviewers for the time and effort they have dedicated to reading our work and for their invaluable insightful comments and constructive criticism that have allowed us to enhance the quality of our manuscript.

Reviewer #1: This manuscript experimentally investigates the parameters that affect the stability of EHD jet printing of poly (ethylene oxide) patterns using solvent-based inks, some guidance is provided for the development and improvement of steady-state printing. Although some results are obtained from experiment and calculation, there are still some issues, we suggest that it be published after major modifications. In addition, there are the following questions that the author should consider and revise:

Reviewer #1: 1. The introduction section needs to be enriched with previous research in EHD area.

Answer: The introduction section of the manuscript was complemented with additional recent references related to EHD jet printing, as suggested by the reviewer.

Reviewer #1: 2. In section 2.3, we suggest that a picture of the device be listed, and that the device be analyzed in conjunction with the picture to facilitate the reader's understanding.

Answer:  Following the reviewer’s suggestion, an image of our printer for electrostatic jet deflection was added in Section 2.3, as a new figure (Figure 1) in the revised manuscript.

Reviewer #1: 3. The sources of all formulas in the manuscript need to be clarified.

Answer:  All the formulas were deduced by the authors, not obtained from any previous manuscript. They are all relatively simple mass/volume balance equations. The meaning of all parameters that appear in the formulas used in the manuscript was defined in the revised version of the manuscript.

Reviewer #1: 4. In section 3.1, the associations of polymer molecular weight with viscosity and conductivity are not clearly articulated. The authors also changed the viscosity when exploring the influence of the polymer molecular weight, and had problems controlling the variables.

Answer:  In Section 3.1, we produced ink batches by dissolving poly (ethylene oxide) of different molecular weights into a solvent composed of water:ethanol (1:1 by weight). We took into account that larger molecular weight will result in more viscous inks if the polymer concentration remained the same (as a result of increased polymer chain entanglements). Therefore, we produced the ink batches by reducing the polymer concentration while increasing the polymer molecular weight, in order to get similar ink viscosity (1.8 – 1.9 Pa s). As we reduced the polymer concentration in the ink while using higher polymer molecular weights, lower quantities of polymer powder were added into the solvent, and consequently the electrical conductivity of the ink was lower, as observed in Table S1 from the supporting information file. We modified the discussion on page 7 to clarify this point.

Reviewer #1: 5. Table S1, table A.1… appears several times in the manuscript, but there is only one table 1 in the article.

Answer: We corrected this mistake. We referred always to “Table S1” from the supporting information file.

Reviewer #1: 6. Figures 4b, c, d need to be revised and the working conditions represented by the black and white squares need to be identified in each figure.

Answer:  Following the reviewer’s suggestion, Figure 4 (in the revised manuscript, Figure 5) was updated by defining a general legend in graph “a”. This legend is the same for graphs “b”, “c” and “d”.

Reviewer #1: 7. What the association of high polymer molecular weight with the drop surface drying? Authors need to elaborate clearly in the text.

Answer:  Because of the higher entanglement number associated with higher molecular weight polymers, at a set polymer weight, an ink based on a polymer with high molecular weight would have a higher viscosity. Thus, when fixing the viscosity, the ink based on the polymer with the higher molecular weight will have a lower amount of polymer and thus it will take longer to dry the surface of its related pendant drop.  In other words, the concentration of the polymer on the drop surface is lower if polymers with high molecular weight are used to produce the ink, minimizing the effect of the drop surface drying during the printing. We clarified this point on page 8.

Reviewer #1: 8. As we all know, the main factors affecting the atomization of conical jets are voltage, flow rate, conductivity and surface tension, why did the authors choose polymer molecular weight for their study?

Answer: The main motivation of this manuscript is improving the stability of EHD jet printing during time in a practical manner. The range of all the parameters listed by the reviewer depend on the ink properties, and particularly the polymer molecular weight. We chose to analyze the effect of the polymer molecular weight to further elaborate on previous theoretical work focusing on process parameters instead of other equally important parameters such as those of the ink (type, amount and molecular weight of polymer, ink viscosity, etc)

Reviewer #1: 9. The logic of the article is confusing and needs to be adjusted to make it reader friendly.

Answer: Following the reviewer’s suggestion, we improved the explanation of the motivation of the work and its structure by introducing a range of clarifying comments within the manuscript. We have extended the abstract with conclusions, and included a description of the article structure at the end of the introduction.

Reviewer #1: 10. Please use updated and recent papers in the literature review to give more sense to the reader.

A conservative level set method for liquid-gas flows with application in liquid jet atomization. Experimental and Computational Multiphase Flow, 5, 67-83 (2023).

Experimental study on electrohydrodynamic atomization (EHDA) in stable cone-jet with middle viscous and low conductive liquid. Experimental Thermal and Fluid Science, 121, 110260.

Answer: Following the reviewer’s suggestion, the bibliography in the revised manuscript was updated considering the recent papers proposed by Reviewer #1 and other additional new works.

Reviewer 2 Report

Comments and Suggestions for Authors

The following are my comments:

(1)This journal is related with the field of the nanomaterials. The authors have to show/discuss some things that can be related with the nanomaterials, such as nanodroplets.

(2) ....150 Hz and 50 Hz...->...150 Hz, and 50 Hz...

(3) The meaning of the parameters  used in Eqs. 1-2 and others equations should be shown.

(4) "Each printing was an 411

independent experiment that started after cleaning the nozzle to get a fresh drop...."This sentence should be revised. 

(5) Is the stability of the EHD printing depended on the polymers used? Why did the authors use  PEO?  If some other polymers are used, how can the stability be expected? The authors may have to discuss this.

(6) The authors mentioned about the measurement of electrical conductivity of Ink in Sec. 2.2. The results seem not showing. 

Comments on the Quality of English Language

Minor editing of English language required

Author Response

We would like to express our gratitude to the reviewers for the time and effort they have dedicated to reading our work and for their invaluable insightful comments and constructive criticism that have allowed us to enhance the quality of our manuscript.

Reviewer #2: The following are my comments:

Reviewer #2: 1. This journal is related with the field of the nanomaterials. The authors have to show/discuss some things that can be related with the nanomaterials, such as nanodroplets.

Answer:  Following the reviewer’s suggestion, in the introduction we specified that EHD jet printing enables producing continuous fibers with diameters in the nanometer scale regime (page 1).

Reviewer #2: 2. ....150 Hz and 50 Hz...->...150 Hz, and 50 Hz...

Answer: We corrected this mistake in the description of Figure 2 (which is Figure 3 in the revised manuscript).

Reviewer #2: 3. The meaning of the parameters  used in Eqs. 1-2 and others equations should be shown.

Answer:  Following the reviewer’s suggestion, the meaning of all parameters in Equations 1-2 was provided in Section 2.4.

Reviewer #2: 4. "Each printing was an independent experiment that started after cleaning the nozzle to get a fresh drop...."This sentence should be revised.

Answer: Following the reviewer’s suggestion, we clarified this sentence in Section 3.3.

Reviewer #2: 5. Is the stability of the EHD printing depended on the polymers used? Why did the authors use  PEO?  If some other polymers are used, how can the stability be expected? The authors may have to discuss this.

Answer: The stability of the EHD jet printing will depend on the polymer used, as the polymer and its concentration will define the ink properties (viscosity, electrical conductivity, surface tension). In this manuscript, we mainly used PEO in our experiments to benefit from its stability in electrohydrodynamic processes. We are confident that the dependencies on ink properties (notably ink viscosity, ink electrical conductivity, and polymer concentration) and operational parameters observed in our work have broader applicability and should therefore be relevant to various polymers (with similarly as good solute-solvent interactions as in our case). Nonetheless, it is important to note that these parameters, such as ink viscosity and electrical conductivity, are highly contingent on the ink formulation, including the polymer used. Distinct polymers and formulations will inevitably lead to divergent ink properties, resulting in varying jet behaviors under identical operational parameters to those employed in our study. It will be necessary, therefore, to establish a proper range of operational parameters for each polymer/solvent combination to ensure the ejection of stable jets. Despite these variations, the trends we have identified could hold true for inks composed of different polymers and solvents. Following the reviewer’s suggestion, we introduced a comment in this direction in the final part of the conclusions section.

Reviewer #2: 6. The authors mentioned about the measurement of electrical conductivity of Ink in Sec. 2.2. The results seem not showing.

Answer: The electrical conductivity of the ink batches used in this manuscript is shown in Table S1 in the Supporting Information file.

Reviewer 3 Report

Comments and Suggestions for Authors

This research explored the factors influencing the stability of electronhydrodynamic (EHD) jet printing when creating poly(ethylene oxide) patterns with solvent-based inks. The study aimed to understand the printing process's evolution by concurrently tracking the drop size, jet ejection point, and jet velocity, which were assessed through the application of a periodic electrostatic deflection. The findings suggest that printing instabilities are linked to variations in the drop size and composition, as well as in the jet's ejection point and velocity. These instabilities are likely tied to solvent evaporation and the consequent drying of the drop's surface. In conclusion, for consistent and reproducible printing, it is essential to use polymers with high molecular weights and solvents that reduce surface drying. Additionally, carefully optimizing the needle voltage is crucial to prevent jet ejection mode instabilities.

The reviewer has recommended the acceptance of this paper for publication in the journal, subject to minor revisions. The comments are as follows:

1. Abstract: The abstract should more clearly articulate the results using numerical data, as the current version presents somewhat indistinct findings.

2. Limitation: The study's limitations need to be discussed in greater detail.

Author Response

We would like to express our gratitude to the reviewers for the time and effort they have dedicated to reading our work and for their invaluable insightful comments and constructive criticism that have allowed us to enhance the quality of our manuscript.

 Reviewer #3: This research explored the factors influencing the stability of electronhydrodynamic (EHD) jet printing when creating poly(ethylene oxide) patterns with solvent-based inks. The study aimed to understand the printing process's evolution by concurrently tracking the drop size, jet ejection point, and jet velocity, which were assessed through the application of a periodic electrostatic deflection. The findings suggest that printing instabilities are linked to variations in the drop size and composition, as well as in the jet's ejection point and velocity. These instabilities are likely tied to solvent evaporation and the consequent drying of the drop's surface. In conclusion, for consistent and reproducible printing, it is essential to use polymers with high molecular weights and solvents that reduce surface drying. Additionally, carefully optimizing the needle voltage is crucial to prevent jet ejection mode instabilities.

The reviewer has recommended the acceptance of this paper for publication in the journal, subject to minor revisions. The comments are as follows:

Reviewer #3: 1. Abstract: The abstract should more clearly articulate the results using numerical data, as the current version presents somewhat indistinct findings.

Answer:  The abstract was updated to report the main conclusions obtained in the manuscript, but we are unable to specify any numeric data that will assure stable printing during long times in a general approach, as the EHD jet printing is affected by many parameters at the same time (humidity, ink properties, nozzle voltage, infusion pump rate, etc).

Reviewer #3: 2. Limitation: The study's limitations need to be discussed in greater detail.

Answer:  Following the reviewer’s suggestion, the limitations of EHD jet printing, related to several ink and process parameters, were discussed all along the manuscript.

Round 2

Reviewer 1 Report

Comments and Suggestions for Authors

Since all the issues have been addressed, this manuscript can be accepted in present version.

Comments on the Quality of English Language

It should be improved.